# Singular Nuclei Segmentation for Automatic *HER2* Quantification Using CISH Whole Slide Images

**DOI:** 10.3390/s22197361

**Published:** 2022-09-28

**Authors:** Md Shakhawat Hossain, M. M. Mahbubul Syeed, Kaniz Fatema, Md Sakir Hossain, Mohammad Faisal Uddin

**Affiliations:** 1Department of CS, American International University-Bangladesh, Dhaka 1229, Bangladesh; 2RIoT Research Center, Independent University, Bangladesh, Dhaka 1229, Bangladesh; 3Department of CSE, Independent University, Bangladesh, Dhaka 1229, Bangladesh

**Keywords:** *HER2* grading, whole slide image, nuclei segmentation, U-net, digital pathology

## Abstract

Human epidermal growth factor receptor 2 (*HER2*) quantification is performed routinely for all breast cancer patients to determine their suitability for *HER2*-targeted therapy. Fluorescence in situ hybridization (FISH) and chromogenic in situ hybridization (CISH) are the US Food and Drug Administration (FDA) approved tests for *HER2* quantification in which at least 20 cancer-affected singular nuclei are quantified for *HER2* grading. CISH is more advantageous than FISH for cost, time and practical usability. In clinical practice, nuclei suitable for *HER2* quantification are selected manually by pathologists which is time-consuming and laborious. Previously, a method was proposed for automatic *HER2* quantification using a support vector machine (SVM) to detect suitable singular nuclei from CISH slides. However, the SVM-based method occasionally failed to detect singular nuclei resulting in inaccurate results. Therefore, it is necessary to develop a robust nuclei detection method for reliable automatic *HER2* quantification. In this paper, we propose a robust U-net-based singular nuclei detection method with complementary color correction and deconvolution adapted for accurate *HER2* grading using CISH whole slide images (WSIs). The efficacy of the proposed method was demonstrated for automatic *HER2* quantification during a comparison with the SVM-based approach.

## 1. Introduction

Breast cancer is one of the world’s most frequent forms of cancer. In the United States alone 268,600 cases were diagnosed among women in 2019, which climbed to 330,840 cases in 2021 [1]. Approximately 20% of the patients are *HER2* positive due to *HER2* gene amplification or subsequent *HER2* protein over-expression [2]. *HER2*, a transmembrane tyrosine kinase receptor encoded by the ERBB2 gene on chromosome 17q12, is a predictive and prognostic biomarker for breast, gastric and other cancers [3]. *HER2* grading is done for all breast cancer patients to identify a *HER2*-positive patient. As an aggressive subgroup, *HER2*-positive breast cancer is treated with anti-*HER2* targeted therapy, such as trastuzumab or lapatinib, to destroy the nucleus of the cancer cell [4,5,6,7,8]. Targeted therapy improves the patient’s condition, and in 1998 the FDA approved trastuzumab to treat *HER2*-positive breast cancer patients. However, if such treatment is given to *HER2*-negative patients, it may cause cardiac toxicity [9]; in addition, it is highly expensive [9,10,11]. Therefore, an accurate *HER2* grading is crucial for designing a treatment plan.

Clinically, *HER2* positivity is determined by counting a myriad of *HER2* genes inside nuclei or subsequent *HER2* proteins outside the nuclei in the cell membrane, as illustrated in Figure 1. Thus, *HER2* quantification methods can be divided into two groups: *HER2* protein based and *HER2*-gene based. Of the two, *HER2* gene-based tests are considered more reliable. Immunohistochemistry (IHC), FISH, and CISH are the FDA approved tests for *HER2* quantification [12,13,14,15]. IHC is a protein-based qualitative test where FISH and CISH count *HER2* genes, and IHC rates the intensity of membranous staining as 0, 1+, 2+, or 3+. However, an IHC test is not conclusive. The American Society of Clinical Oncology and College of American Pathologists (ASCO/CAP) recommend conducting a reflex FISH or CISH test to confirm the *HER2* grade [12].

For FISH or CISH analysis, the invasive breast cancer regions are first identified from a biopsy. After that, singular nuclei suitable for *HER2* quantification are selected. A singular nucleus that is not overlapped with another nucleus and does not have any missing parts is suitable for quantification, as shown in Figure 2. Usually, a healthy nucleus has two copies of CEP17 and four copies of the *HER2* gene. The copy of the *HER2* gene increases in comparison to the copy of the CEP17 gene in a *HER2*-positive nucleus. Therefore, *HER2* and CEP17 signals are counted for singular nuclei from cancer regions. As seen in Figure 3, the inclusion of non-singular nuclei in the quantification causes inaccurate signal counting and incorrect analysis. As a result, quantifying only the singular nuclei is a must for precise *HER2* grading. ASCO/CAP recommends quantifying at least 20 nuclei. Then, the *HER2* grade is determined based on the *HER2*-to-CEP17 ratio and the average *HER2* copy as shown in Table 1.

For the *HER2* gene-based quantification, laboratories select suitable nuclei and then count signals manually from FISH or CISH slides under a microscope. It is a labor-intensive and time-consuming task that is also vulnerable to subjective interpretation. As a result, an automated quantification method has many advantages. Several methods have been proposed for counting signals from FISH in a semi-automated or automated approach [16,17,18,19]. A few methods were proposed to quantify CISH slides automatically [20]. The choice of FISH versus CISH varies among institutions. FISH uses fluorescence imaging and the tests require special training and setup for the test. FISH dyes are expensive, and preparing a specimen takes a long time. On the other side, CISH uses bright-field imaging and does not require any special setup or training. Plus, the CISH dyes are cheaper, and the specimen preparation time is shorter. Thus, the CISH test is more practical than FISH. Previously, an automated *HER2* grading system called Shimaris-PACQ was proposed using CISH WSI by Yagi et al. [20]. CISH used SVM to detect singular nuclei, and the system was considered the state of the art for automatic *HER2* quantification. However, it failed to detect singular nuclei on some occasions, which led to inaccurate results. Therefore, in this paper we propose a robust nuclei detection method using deep learning for reliable automatic *HER2* grading using CISH.

## 2. Literature Review

Cell or nuclei-based assessment is a widely used technique in biomedical image analysis for a variety of purposes, including determining cancer grade, counting bio-marker signals inside a nucleus, distinguishing cancerous nuclei from non-cancerous nuclei, nuclei characterization, assessing tumor cellularity [19,20,21,22,23,24,25,26,27,28,29,30,31,32,33,34,35,36,37,38]. These assessment methods can be divided into morphological and molecular image analysis. Hematoxylin & Eosin (H&E) is the most commonly used staining method for morphological analysis of features such as nuclei shape, size, and distortion. Molecular analysis is used to detect and quantify molecules that are not present in H&E specimens. Popular techniques related to this research include CISH and FISH. As a result, we concentrated primarily on the nuclei segmentation methods developed for CISH and FISH specimens. Several methods have been proposed for segmenting nuclei from FISH slides [19,21,37,38]. On the other side, only a limited number of methods are available for segmenting nuclei from CISH slides [20].

One of the major challenges in analyzing histopathology specimens such as H&E, CISH and FISH is that they preserve the original tissue structure and the nuclei are often part of these structures. The presence of different tissue structures, varied staining and nuclei overlap make nuclei detection challenging. Therefore, a method that has been optimized for one stain and specimen doesn’t work as well for another. Furthermore, because the methods do not apply to the same dataset, it is a challenge to compare their performance. The majority of nuclei detection methods developed for H&E or IHC slides do not work well for CISH and FISH [22,32,33,35]. Plus, the nuclei detection result produced by these methods is not suitable for *HER2* quantification because detecting nuclei is not enough; only singular nuclei are included in the quantification. Furthermore, a method developed for FISH specimens doesn’t generalize well for CISH specimens. Therefore, it is necessary to develop a nuclei detection method for *HER2* quantification using CISH. Yagi et al. proposed a method of detecting singular nuclei from CISH slides, but it failed in some cases when the stain condition changed and the slides were scanned with a different scanner [20]. A commercial application was developed by 3dhistech but it led to a high number of false positives and included non-singular nuclei for quantification. This signifies the importance of a robust singular nuclei segmentation method for CISH slides to perform automatic *HER2* quantification.

The approaches used for nuclei segmentation can be divided into three groups: (1) using image analysis tools such as ImageJ, CellProfiler, 3dhistech and CaseViewer; (2) using traditional machine learning such as SVM and Random Forest and (3) using deep learning such as U-net and other pixel-wise classification methods. Among the three approaches, deep learning has achieved higher accuracy and reliability whereas U-net based methods are leading the way. U-net based pixel-wise segmentation has been found to be the most effective, fast and state of the art [21]. This motivated us to use the U-net-based method for singular nuclei segmentation from CISH slides. For this purpose, we modified the original U-net model to differentiate the boundary pixels inside the nuclei from outside nuclei pixels. To segment singular nuclei, the method was trained on an expert’s manual annotation. Moreover, expert pathologists evaluated the proposed method, and then we compared the results to the SVM-based method. The proposed method outperformed it in a demonstration and was found to be robust against multiple scanners and varying stain conditions. Furthermore, the method was found effective for FISH slides when demonstrated.

## 3. Materials and Methods

The existing state of the art of CISH-based automatic *HER2* quantification failed to segment suitable nuclei in some cases. Accurate marking of the nucleus boundary is important because the system uses the boundary to count the bio-marker signals. If a signal is inside the nucleus or lies entirely on the boundary then it is counted for that nucleus. But if the signals partially overlap with the boundary, they are excluded. The previous, SVM-based nucleus detection method used color deconvolution to separate the nuclei-dye channel, from which the nuclei were detected. This step is very useful when there is cross-talk among dyes in an image like CISH. However, it used intensity values to distinguish noise pixels from nuclear pixels and to mark the boundary, which is not effective if the staining condition of the specimen varies. Plus, this method was highly parameterized and its performance depended on the careful selection of parameters. To develop a more robust nucleus detection method, we relied on deep learning, which allows the extraction and selection of non-handcrafted features by a convoluted neural network (CNN).

Algorithm 1 explains the proposed nuclei segmentation approach, which begins by assessing the quality of the CISH WSI. Automated nuclei detection fails if the quality of the image is not satisfactory, as shown in Figure 4. Figure 5 shows how image quality affects nuclei segmentation. Therefore, before segmenting nuclei, we evaluated the WSI’s quality using the referenceless method proposed by Yamaguchi et al. [39,40].

If the quality of the WSI is satisfactory, only then is it used for nuclei detection. After a quality check, the color of the input image is corrected by comparing it to an ideally stained image. Then, the nucleus dye channel of the image is obtained using color deconvolution. The image is a single-channel gray-scale image used by U-net to segment singular nuclei. Each step in the nuclei segmentation method is explained in detail below.
**Algorithm 1:** Singular nuclei segmentation methodInitialization of $W_{th},Q_{th},\alpha ,\beta ,\gamma ,M,\theta ,P_{Nuclei}^{Ref},P_{CEP17}^{Ref},P_{HER2}^{Ref}$**procedure**SingularNucleiSegmentation($I_{RGB}$,$W_{th}$, $Q_{th}$)    **while** $I_{RGB}!=NIL$ **do**        $W_{i}$ = WhiteCheck($I_{RGB}$)        **if** $W_{i}$ < $W_{th}$ **then**           $Q_{i}$ = QualityCheck($I_{RGB},\alpha ,\beta ,\gamma$)           **if** $Q_{i}\leq Q_{th}$ **then**               $I_{CC}$ = ColorCorrection($I_{RGB},P_{Nuclei}^{Ref},P_{CEP17}^{Ref},P_{HER2}^{Ref}$)               $I_{CDN}$ = ColorDeconvolution($I_{CC},M$)               $I_{Out}$, $Score_{nuclei}$ = Segmentation($I_{CDN}$, $u_{\theta }$)           **end if**        **end if**    **end while**    return $I_{Out}$, $Score_{nuclei}$**end procedure****procedure**WhiteCheck($I_{RGB}$)    $I_{gray}$ = 0.299×$I_{R}$ + 0.587×$I_{G}$ + 0.114×$I_{B}$    **while** $Pixel_{I_{gray}}!=NIL$ **do**        **if** $Pixel_{I_{gray}}\geq 200$ **then**           $W_{count}$++        **end if**    **end while**    $W_{pixels}=W_{count}/Pixel_{I_{gray}}$    return $W_{pixels}$**end procedure****procedure**QualityCheck($I_{RGB},\alpha ,\beta ,\gamma$)    *Q* = α+β×Blurriness+γ×Noise    Blurriness and Noise are calculated using Equations (Equation 1) and (Equation 2)    return *Q***end procedure****procedure**ColorCorrection($I_{RGB},P_{Nuclei}^{Ref},P_{CEP17}^{Ref},P_{HER2}^{Ref}$)    **while** $Pixel_{I_{RGB}}!=NIL$ **do**        For each pixel $Pixel_{I_{RGB}}$ of $I_{RGB}$, estimate $I_{R}^{C},I_{G}^{C},I_{B}^{C}$ using Equations (Equation 4) and (Equation 5)    **end while**    Construct $I_{CC}$ from $I_{R}^{C},I_{G}^{C},I_{B}^{C}$    return $I_{CC}$**end procedure****procedure**ColorDeconvolution($I_{CC},M$)    Camera response $I_{CC_{k}}$ is estimated using Equation (Equation 7)    $a_{Nuclei}=I_{CC_{k}}\cdot M_{Nuclei}$  ; k=R,G,B    return $I_{CDN}\leftarrow a_{Nuclei}$**end procedure****procedure**Segmentation($I_{CDN}$, $u_{\theta }$)    $I_{Seg}=u_{\theta }\left(I_{CDN}\right)$    $I_{Out}$ = PostProcess($I_{Seg}$)    return $I_{Out}$**end procedure****procedure**PostProcess($I_{Seg}$)    $I_{Seg}=$ Apply Morphological Opening on $I_{Seg}$    Count nuclei in $I_{Seg}$    **while** Nuclei!=NIL **do**        Calculate Circularity and Area of each Nuclei using Equation (Equation 9)        **if** Circularity>0.80||(Area>500&&Area<5000) **then**           $Score_{nuclei}=\left(Circularity+Curvature\right)/2$           $I_{PP}=I_{PP}\;\&\;Nuclei$        **end if**    **end while**    return $I_{PP}$, $Score_{nuclei}$**end procedure**

### 3.1. Dataset

In this experiment, we used 32 randomly selected breast cancer CISH WSI specimens. The CISH WSIs were scanned using a 3dhistech WSI scanner with 40× objective lens (NA 0.95) which provided an image resolution of 0.13 µm/pixel. The specimens were de-identified and did not contain any details of the patient. However, for confidentiality reasons, the dataset cannot yet be made public, but we did select and export images from the WSI using 3dhistech CaseViewer. For training the proposed U-net model, we used a set of 35 images exported from 22 CISH WSIs. Another set of 15 images exported from 10 CISH WSIs were used for testing the model. The test dataset was unseen in the training and included the cases for which the previously proposed SVM-based method failed to detect sufficient singular nuclei.

### 3.2. Image Quality Assessment

Image quality evaluation methods can be broadly categorized into three groups: (1) full reference-based assessment (FR-IQA), (2) reduced reference assessment (RR-IQA) and (3) no reference or referenceless assessment (NR-IQA). A full reference assessment method evaluated the quality by comparing with it a reference, considered to be the ideal image. Reduced-reference assessment evaluates the perceptual quality of an image through partial information of the corresponding reference image. The goal of the no-reference method is to estimate the perceptual image quality in accordance with subjective evaluations without using any reference. This approach is suitable when it is difficult to obtain an ideal reference images as in our case.

In our work, we used the no-reference quality evaluation method proposed by Yamaguchi et al. [40] to evaluate the quality of a WSI for automated image analysis and diagnosis. This method first estimates the number of white pixels in the input image, $I_{RGB}$. A pixel, $Pixel_{I_{gray}}$ with an intensity value higher than 200 in grayscale is considered white. If an image contains white pixels $W_{pixels}$ more than a threshold of $W_{th}$, say 50%, then it is considered useless for analysis and rejected for nucleus segmentation as it doesn’t contain enough tissue. The image was converted to grayscale considering the human sensitivity to red $I_{R}$, green $I_{G}$ and blue $I_{B}$ color. After that, the quality of the image was estimated based on its blurriness and noise. If these indices of an image are high, it is considered to be poor quality; thus, it was rejected for nuclei segmentation when the quality was higher the selected threshold $Q_{th}$.

The difference between the local maxima and minima is calculated as the width of the edges. After that, the average width for the edges is calculated, which serves as the quality index. Blurry edges have small gradients that result in large width values compared to sharp edges. Blurriness is the average width of edges as shown in Equation (Equation 1). A edge is defined by its gradient, which is higher than a pre-defined threshold value. The edge width is obtained by measuring the distance between the local maximum and local minimum of edge pixels. Then, the total width of all edges is divided by the number of edges which gave the blurriness index. A blurry image has a small gradient on the edges resulting in higher width for the edges. Thus, a large average width indicates a blurry image in contrast to a smaller, sharp image.
(1)$$\begin{array}{c} Blurriness=\frac{1}{E}\sum_{i=1}^{E}w\left(i\right) \end{array}$$
where *E* is the number of total edges, and w(i) is the width of edge *i*. A pixel is considered noise if its value is high and independent of its surrounding pixels. First, high-intensity pixels were detected using the unsharp mask and were either noise or edges. Then, the minimum difference between the center pixel and surrounding pixels in a 3 × 3 pixel window is calculated at all pixels to remove the edge pixels. After that, the average value of these minimum differences was calculated to derive the image noise. A higher value indicates more noise.
(2)$$\begin{array}{c} Noise=\frac{1}{N}\sum_{i=1}^{N}{\left[d_{min}\left(i\right)\right]}^{2} \end{array}$$
where *N* is the number of total pixels, and $d_{min}\left(i\right)$ is the minimum difference for pixel *i*. Finally, the quality degradation index was estimated using linear regression analysis in which blurriness and noise were used the predictors, and the co-efficients of predictors were derived by training the regression model given in Equation (Equation 3). The mean square error (MSE) between the original images and the digitally degraded versions was used in place of the quality degradation index to train the model. In our experiment, we found a linear relationship between the MSE and quality degradation indices.
(3)$$\begin{array}{c} Quality\;degradation\;index=\alpha +\beta Blurriness+\gamma Noise \end{array}$$

Here, α is the intercept and β and γ are the co-efficients of predictors.

### 3.3. Color Correction

Color correction is a typical step in pathological image analysis to handle the color variations that may be caused by a variety of factors, including staining conditions, WSI scanner settings and the WSI viewer. The majority of color correction techniques rely on data from an external reference which is considered to be an ideally prepared specimen. The proposed method modifies the color distribution of the input image using a reference CISH WSI for nuclei segmentation using the reference-based color correction method of Murakami et al. [41], which states that the values of a pixel can be derived by multiplying the primary vectors with some weights. Figure 6 shows the color distribution of a CISH image.

Thus, the color of a pixel as a CISH image can be represented using the following model:(4)$$\left[\begin{array}{c} R\\ G\\ B \end{array}\right]=\left[\begin{array}{ccc} P_{Nuclei} & P_{CEP17} & P_{HER2} \end{array}\right]\left[\begin{array}{c} W_{1}\\ W_{2}\\ W_{3} \end{array}\right]$$
where R,G,B are the red, green and blue values of a pixel in RGB channels; $P_{Nuclei}$, $P_{CEP17}$, $P_{HER2}$ are the primary color vectors; and $W_{1},W_{2},W_{3}$ are the weighting coefficients. The primary vectors are derived from the image. For example, the primary vector $P_{Nuclei}$ is derived by calculating the average red, green and blue values from the nuclei only areas. Similarly, the $P_{CEP17}$ and $P_{HER2}$ vectors are derived from the CEP17 and *HER2* only areas. The proposed method used the model (Equation 1) to correct the color of an input image based on the given in Equation (Equation 2):(5)$$\left[\begin{array}{c} R^{C}\\ G^{C}\\ B^{C} \end{array}\right]=\left[\begin{array}{ccc} P_{Nuclei}^{Ref} & P_{CEP17}^{Ref} & P_{HER2}^{Ref} \end{array}\right]\left[\begin{array}{c} W_{1}\\ W_{2}\\ W_{3} \end{array}\right]$$
where $R^{C}G^{C}B^{C}$ are the color-corrected values of the pixel; $P_{Nuclei}^{Ref}$, $P_{CEP17}^{Ref}$, $P_{HER2}^{Ref}$ are the reference color vectors; and $W_{1},W_{2},W_{3}$ are the weights. The reference vectors $P_{Nuclei}^{Ref}$, $P_{CEP17}^{Ref}$, and $P_{HER2}^{Ref}$ are derived from the nuclei, CEP17 and *HER2* only areas of the reference image. The values of the weighting coefficients are derived by inverting the equation, shown in model (Equation 1). Then the color-corrected image is used for color deconvolution where the nuclei-dye channel is separated which is for the nuclei segmentation.

### 3.4. Color Deconvolution

Color deconvolution is useful for separating the dye contribution if the cross-talk of dyes is significant in the specimen. In the case of CISH, three different dyes were used: blue for highlighting the nuclei, magenta for CEP17 and black for *HER2*. Based on the Beer–Lambert law, the proposed method applied color deconvolution to the CISH image to separate the nuclear dye channel image which was then used to segment the nuclei using U-net. The cross-talk of dyes is not significant in the FISH image. The Beer-Lambert law is the linear relationship between absorbance and concentration of an absorbing species. For an imaging device it can be represented using the following mathematical model:(6)$$\begin{array}{cc} \; & g=Ma\\ \Rightarrow & a=gM^{-1} \end{array}$$
where *g* is the camera response; *M* is the color matrix; and *a* is the dye contribution. Using this model, we derived the contribution of each dye. Now, the camera response *g* can be derived from the input image as:(7)$$g_{k}=\log\frac{I_{ok}}{I_{k}};k=R,G,B$$
where $g=(g_{R}\;$$g_{G}\;$$g_{B}{)}^{T}$ is the optical density; $I=(I_{R}\;$$I_{G}\;$$I_{B}{)}^{T}$ is the intensity of RGB components of every pixel. $I_{o}=(I_{oR}\;$$I_{oG}\;$$I_{oB}{)}^{T}$ is the average intensity of glass pixels. The color matrix was composed of an optical density vector for specific colors. Therefore, the *M* was derived based on the average *R*, *G* and *B* values of nuclei only, CEP17 only and *HER2* only areas as
(8)$$M=\left[\begin{array}{ccc} AvgR_{Nuclei} & AvgG_{Nuclei} & AvgB_{Nuclei}\\ AvgR_{CEP17} & AvgG_{CEP17} & AvgB_{CEP17}\\ AvgR_{HER2} & AvgG_{HER2} & AvgB_{HER2} \end{array}\right]$$

Here each element of *M* represents an optical density derived by dividing by the glass intensity and then performing the log. The nuclei, CEP17 and *HER2* color responses are denoted as $M_{Nuclei},M_{CEP17}$ and $M_{HER2}$. Thus, the model in (Equation 6) results in three stain-separated grayscale images $\left(a_{Nuclei},a_{CEP17},a_{HER2}\right)$, belongs to nuclei, CEP17 and *HER2*. The proposed method uses the nucleus channel for U-net.

### 3.5. U-Net for Nuclei Segmentation

The proposed method uses a U-net [28] network to detect the untruncated and non-overlapped singular nuclei from a grayscale image. Since the conventional U-net is a semantic segmentation, plural nuclei are sometimes included in the segmentation result and difficult to separate them. To avoid this problem by an approach similar to instant segmentation, we trained the U-net with 3-classes; background, boundary, or inside the nuclei, respectively [32]. Gray-scale images obtained by applying color deconvolution to the CISH images were used to train the network. The output of the nuclei detection model has 3 channels, each having the same height and width as the input image. Their pixel values represent the probability of each pixel being background, boundary, or inside the nuclei. A pixel belonging to the boundary class means that it is on or inside an annotated boundary within 2 pixels. We trained the U-net based segmentation model $u_{\theta }:I_{CDN}\to S$ such that the segmentation of the nuclei dye image $I_{CDN}$ can be obtained as $I_{Seg}=u_{\theta }$ ($I_{CDN}$) where $u_{\theta }$ is a non-linear function; θ is a vector of parameters; and *S* is the set of segments. The parameter vector θ is derived from the training for which the accuracy of the segmentation model $u_{\theta }$($I_{CDN}$) is minimal. If *L* is a segment of $I_{CDN}$, then L is represented as $L=\phi \left(I_{Seg}\right)$ where ϕ is the labeling operator and $I_{Seg}$ is a segmentation approximation of $I_{CDN}$.

We used 35 color-deconvoluted nuclei dye images (384 × 768 × 1) for training collected from the CISH WSI of breast cancer patients. Each image contained approximately 100 nuclei and a total of 3500 were annotated manually for training. Figure 7 shows the pathologist’s annotation from which the nuclei boundary (NB) label was produced and served as the ground truth. Data were augmented by applying vertical flip, horizontal flip and random zooming (×1.0 ×1.1) during training. The network was trained by the Adam optimizer and the loss function was categorical cross-entropy. Figure 8 illustrates the training process. The epoch size was 30 and the learning rate was 0.0001. SoftMax was used as the output function. We also used batch normalization. The U-net network consisted of a couple of encoding and decoding layers. Another set of 15 images was prepared for testing the model including cases where the SVM model failed. Figure 9 illustrates the process of predicting nuclei for a given input image and the post-processing for the prediction. In post-processing, the inside class map is transformed into a binary map. The inside region is marked in blue in the prediction rectangle in Figure 9. To recover the shape, we simply applied a dilation operation at the end of segmentation to each connected component.

### 3.6. Scoring Nuclei Suitability

A non-singular nucleus tends to have low circularity compared to a singular nucleus. Again, if some parts are missing then the size of the nucleus becomes very small. On the other hand, if multiple nuclei are overlapped then its area becomes larger compared to that of a singular nucleus. Therefore, the proposed method scored each segmented nucleus based on it circularity, which was estimated as
(9)$$\begin{array}{cc} Circularity & =\frac{4\pi Area}{Perimeter^{2}} \end{array}$$

The proposed method eliminates a nucleus if its circularity<0.80 or area<500 or area>5000. The rest of the nuclei were assigned a score based on its circularity.

## 4. Results

### 4.1. Accuracy of Proposed Nuclei Segmentation

The proposed method was demonstrated on 14 breast cancer cases by two different scanners that included 7 CISH, for which the SVM-based method failed, and 7 FISH WSIs. First, the performance of the proposed method was evaluated for 7 CISH WSIs using the intersection over union (IoU) metric. IoU measures the number of common pixels between the pathologist’s annotation and the model’s prediction divided by the total number of pixels covered by both, as shown in Equation (Equation 7).
(10)$$IoU=\frac{Annotation\cap Prediction}{Annotation\cup Prediction}$$

However, when judging the performance of the nuclei detection, the IoU metric could be misleading. Such an example is illustrated in Figure 10. Both segmentations resulted in a similar IoU, but the left one was completely useless for singular nuclei quantification. Moreover, the inclusion of a non-nuclear area or the exclusion of some part of the nucleus could affect signal counting even if the difference is very small. Therefore, we relied on the pathologist’s manual evaluation to identify the false and true positives. For the *HER2* quantification, we needed to use only a limited number of singular nuclei, thus, the number of false negatives was not a major issue as long as the method detected enough singular nuclei. Two pathologists marked the true positives and false positives for 7 CISH cases where the SVM-based method failed. Then we compared the SVM and the proposed U-net nuclei detection provided in Table 2, which shows that the proposed method increased true positive detection and reduced false detection significantly. The true positive rate and false positive rate were 0.96 and 0.03 for U-net while it was 0.60 and 0.39 for SVM, accordingly. Figure 11 shows nuclei detection by the SVM and U-net-based methods for the same image.

### 4.2. Application for Automatic *HER2* Quantification

We also applied the proposed method to identify singular nuclei using FISH, but used the intensity image instead of the dye image. To evaluate the efficacy of the proposed method for automatic *HER2* quantification method, we integrated it with the Shimaris–PACQ. Then, we compared the results with the pathologist’s manual CISH and FISH, counts and automated CISH using SVM-nuclei detection as shown in Figure 12 and Figure 13. From the figures, it is clear that the proposed method yielded higher concordance for automatic *HER2* quantification with the pathologist’s manual quantification compared to SVM-based automated quantification. In practice, manual FISH counts werew considered the clinical guideline. The correlation between the pathologist’s FISH count and the Shimaris–PACQ was 0.99 when used with the proposed method, while it was 0.29 when used with SVM-based method.

The proposed method segmented singular nuclei where the SVM-based method failed. The method was integrated with the Shimaris–PACQ to ensure its effectiveness for automated quantification. Shimaris–PACQ achieved higher concordance with pathologist’s results when used with the proposed method compared to SVM. Thus, it can be stated that the proposed nuclei segmentation method enabled precise and reliable automatic *HER2* quantification. Moreover, this method separated singular nuclei from FISH. This method is also effective for FISH-based automatic quantification, as shown in Figure 12 and Figure 13. Thus, the proposed nuclei segmentation method is robust regardless of the scanners, staining conditions and histology specimen types.

### 4.3. Time Requirement Analysis

We analyzed the time requirements for the proposed nuclei segmentation method to ensure its practical usability for automatic quantification, as shown in Table 3. We also estimated the time requirement for automatic *HER2* quantification using the proposed nuclei segmentation. Then we compared the results with the time requirement of the previous system that used SVM for nuclei detection [20]. The time was estimated by using a personal notebook with a 2.6 GHz Intel Core i5 processor without an external graphics card. The turnaround time ranged from 1.33 to 4.00 min for the 7 cases, which was 2.90 to 7.10 min in our experiment. Thus, it can be concluded that the proposed nuclei segmentation method saves a significant amount of time for *HER2* assessment compared to the previously proposed automatic CISH–*HER2* quantification.

## 5. Discussion

The proposed nuclei segmentation method segments singular nuclei suitable for *HER2* quantification for breast cancer patients. When trained on a limited dataset, this method produced very few false positives and detected a large number of true positives of singular nuclei, which is a significant improvement over previous methods. In the demonstration, this method outperformed the state-of-the-art [20]. The method [20] for automatic *HER2* quantification using CISH was validated by comparing the results to the pathologists’ manual FISH and manual CISH counts. However, when the staining condition changed and the specimen was scanned with a different scanner, it failed to segment suitable singular nuclei for some CISH cases. The proposed nuclei segmentation method is robust against the stain variation and multiple scanners. Moreover, this method yielded higher concordance with the automatic *HER2* quantification using CISH when compared with pathologists’ FISH and CISH counts. On top of that, the method is applicable for *HER2* quantification using FISH. One significant benefit of having a nuclei segmentation method like the proposed method is that it allows laboratories to choose CISH or FISH based on their convenience.

The [20] method was highly dependent on a large number of parameters, which limited its generalizability. As a result, it failed when the optimal scanner and staining profile conditions changed. The proposed method is less parameterized and has been found to be effective for a variety of scanner and staining conditions. Furthermore, it works for FISH slides that had been scanned by a different scanner with different settings. This ensures that the proposed method is generalizable. When the [20] method does not apply to FISH slides due to its selection of parameters optimized for CISH images, particularly the method’s noise removal technique.

The [30] method increased the average IoU score for nuclei segmentation, but this improvement had no effect on *HER2* quantification because the result was calculated for the overall nuclei pixel segmentation, which included many non-singular nuclei. Furthermore, it frequently misclassified the boundary pixels, which is critical for *HER2* quantification. The [31] method achieved a good segmentation result for FISH images. However its performance in segmenting singular nuclei from the CISH images where the amount of nuclei overlapping was higher than for FISH. Our method not only improved segmentation performance but also ensured its clinical relevance by evaluating its results by experts and combining it with the quantification methods. Most of the nuclei detection methods previously proposed for H&E, CISH and FISH failed when the quality of the image became poor. Image quality is a prerequisite for autonomous image analysis. According to [20], automatic *HER2* quantification systems fail if the image quality is poor. As a result, we used an evaluation method to ensure that only images of sufficient quality were used for nuclei segmentation.

In this paper, we proposed a more robust nuclei detection method based on deep learning. Currently, a major limitation of applying deep learning to medical images is obtaining the training data, which is time-consuming, costly and laborious. The proposed U-net based nuclei detection method worked well with limited training data. This method demonstrated high reliability when trained on a limited dataset. This is an important feature, especially for critical clinical applications where large image samples are difficult to obtain. More training data would improve the accuracy as more nuclei features could be obtained, but the current performance of the model was sufficient to quantify the limited number of nuclei for *HER2* assessment.

Another notable aspect of the proposed method is that it has been tested and found to be effective for multi-modal images. Furthermore, its efficacy was demonstrated through integration and demonstration with both *HER2* quantification systems, CISH and FISH. The proposed nuclei segmentation method was demonstrated with the *HER2* quantification using a personal notebook without a GPU which took a maximum of 4 minutes per case. Time is another advantage of the proposed method. Its practical usability and time requirement are efficient for automatic *HER2* quantification. It is impractical to allocate advanced computing resources in hospitals.

This nuclei segmentation method can be demonstrated for other nuclei-based assessment applications such as the tumor cellularity of breast cancer patients and imaging modalities such as H&E, by optimizing some parameters.

## 6. Conclusions

In this paper, we presented a U-net-based singular-nuclei segmentation method for automatic *HER2* quantification using CISH. Furthermore, the application of the method was proven effective for FISH. It started by assessing image quality, then the image’s color was adjusted to handle the color variation. The method then separated the nuclei dye channel using color deconvolution. These three procedures were used to ensure robustness against stain and scanner variation. The singular nuclei were then segmented using the U-net, which identified the nucleus boundary concurrently and performed well when trained with a small number of images. 

## Figures and Tables

**Figure 1 sensors-22-07361-f001:**
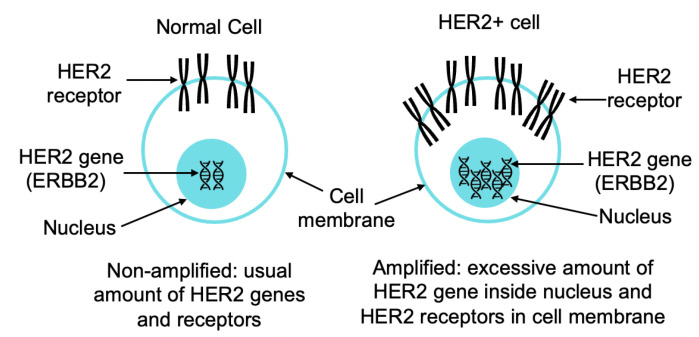
Example of normal and *HER2* positive cell.

**Figure 2 sensors-22-07361-f002:**
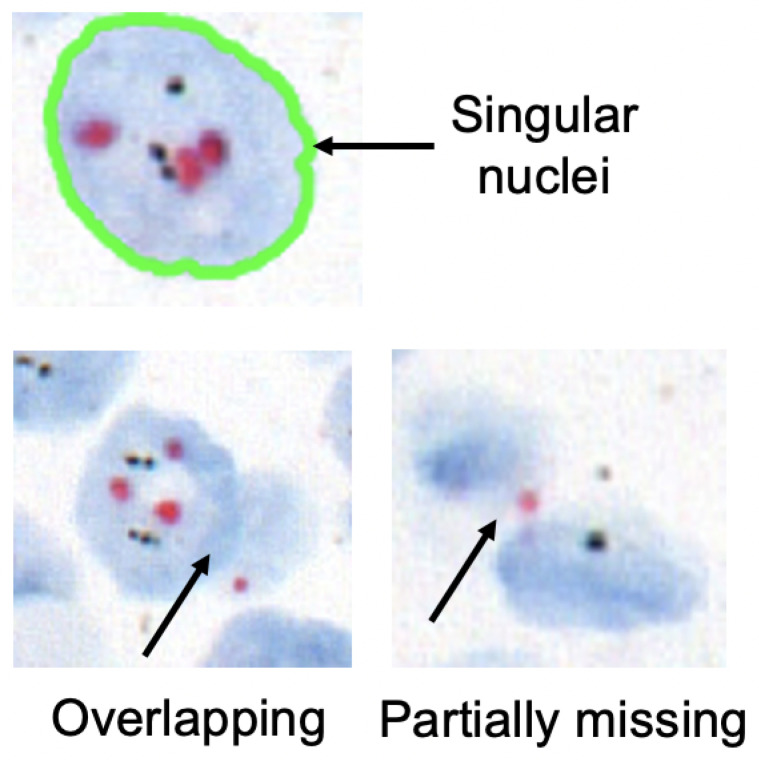
Examples of singular and non-singular nuclei.

**Figure 3 sensors-22-07361-f003:**
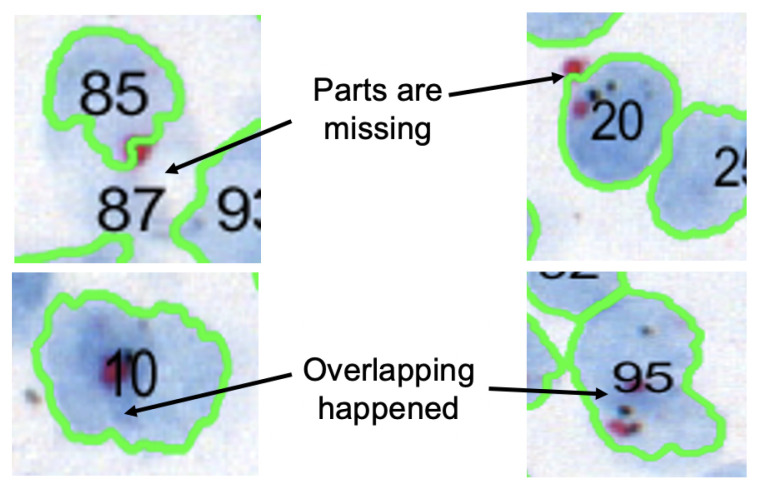
False positives of singular nuclei which may lead to inaccurate signal counting.

**Figure 4 sensors-22-07361-f004:**
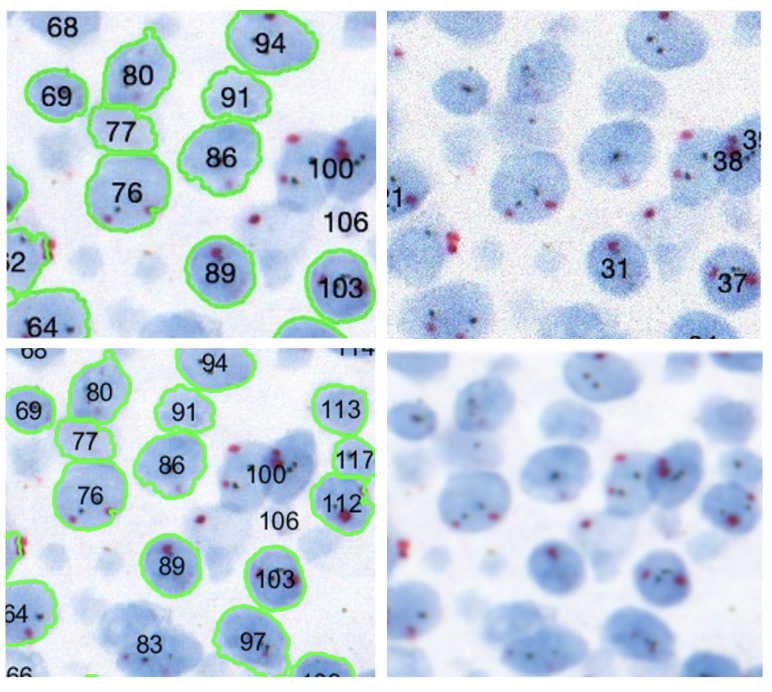
The nuclei detection method succeeded for the original (**top-left** and **bottom-left**) images but failed for the blurry (**bottom-right**) and noisy version of the images (**top-right**).

**Figure 5 sensors-22-07361-f005:**
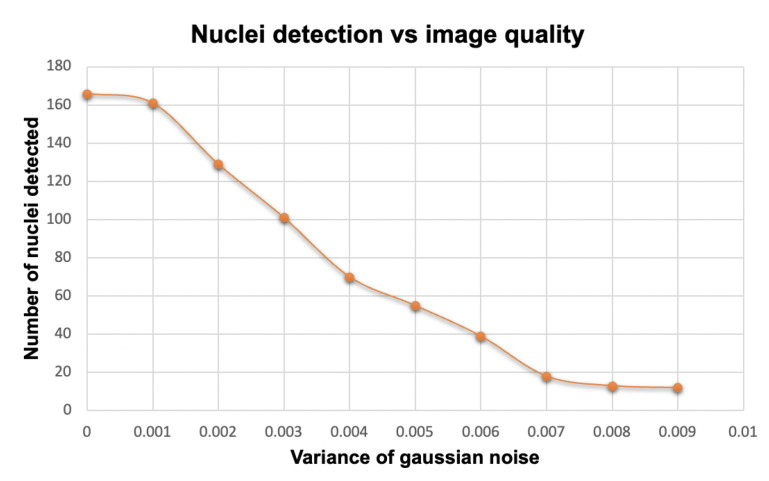
Effect of image quality on the nuclei detection method.

**Figure 6 sensors-22-07361-f006:**
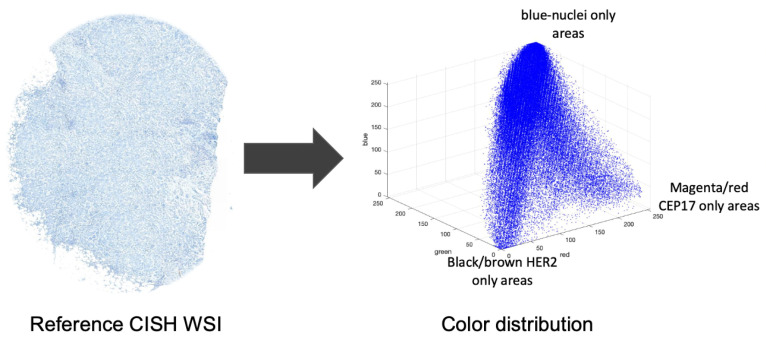
An example model of color distribution of CISH WSI.

**Figure 7 sensors-22-07361-f007:**
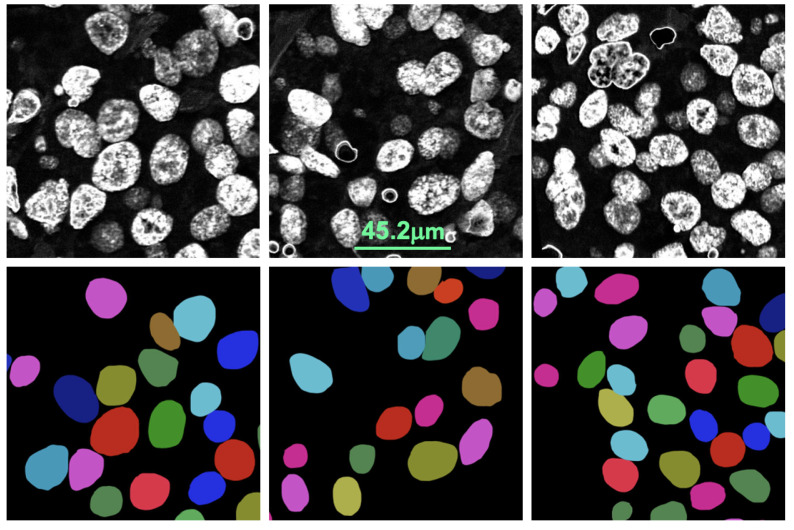
Examples of how the data were annotated. The top row shows the deconvoluted version of the original color images, and the bottom row shows the pathologist’s annotations for the corresponding images.

**Figure 8 sensors-22-07361-f008:**
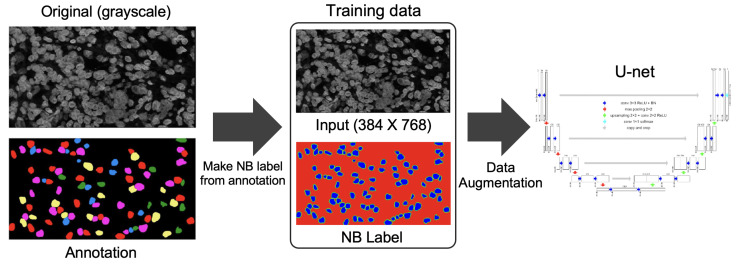
Overview of the training process.

**Figure 9 sensors-22-07361-f009:**
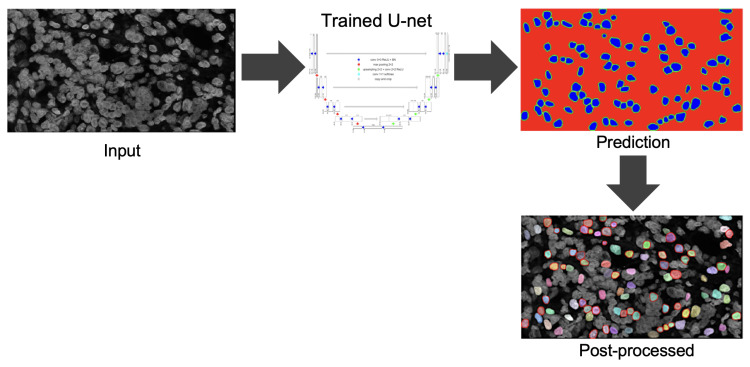
Overview of nuclei segmentation.

**Figure 10 sensors-22-07361-f010:**
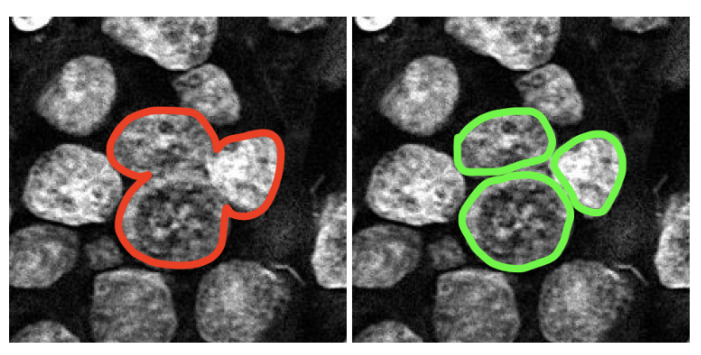
Wrong prediction (**left**) and correct prediction (**right**).

**Figure 11 sensors-22-07361-f011:**
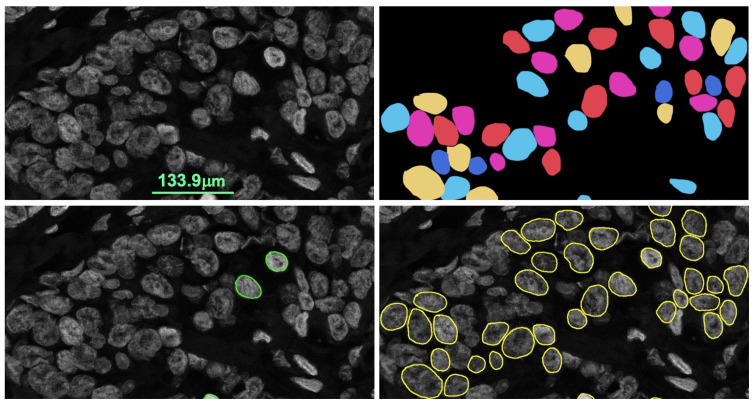
Proposed method outperformed the SVM based nuclei detection. Original image (**top left**), ground truth (**top right**), SVM detection (**bottom left**) and proposed U net segmentation (**bottom-right**).

**Figure 12 sensors-22-07361-f012:**
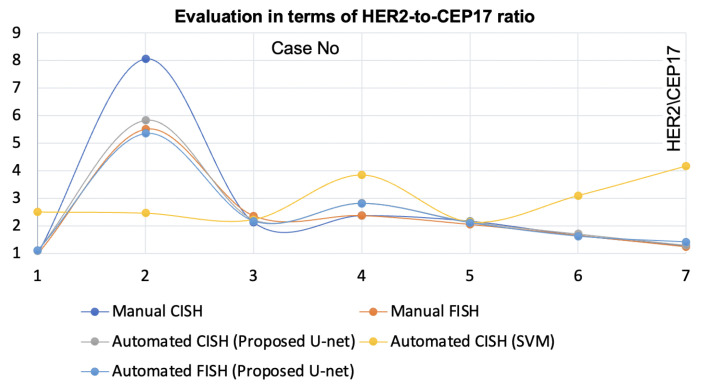
Evaluation of proposed nuclei segmentation enabled automatic *HER2* quantification using both CISH and FISH with respect to the *HER2*––CEP17 ratio.

**Figure 13 sensors-22-07361-f013:**
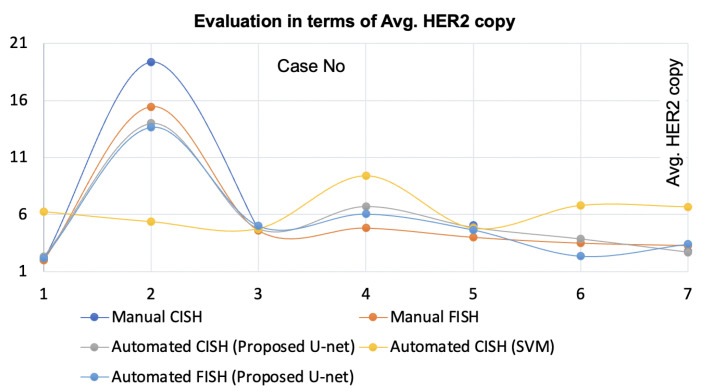
Evaluation of proposed nuclei segmentation-enabled automatic *HER2* quantification using both CISH and FISH with respect to the average *HER2* copy.

**Table 1 sensors-22-07361-t001:** *HER2* grading based on *HER2* and CEP17 counts.

*HER2* Groups	Criteria	*HER2* Status
Group 1	HER2/CEP17≥2 Avg. HER2 copy ≥4	Positive
Group 2	HER2/CEP17≥2 Avg. HER2 copy <4	Positive
Group 3	HER2/CEP17<2 Avg. HER2 copy ≥6	Positive
Group 4	HER2/CEP17<2 Avg. HER2 copy ≥4 & <6	Equivocal
Group 5	HER2/CEP17<2 Avg. HER2 copy <4	Negative

**Table 2 sensors-22-07361-t002:** Comparison of SVM and Proposed U-net based nuclei detection results.

	SVM			Proposed U-Net		
**SI**	**TP**	**FP**	**IoU**	**TP**	**FP**	**IoU**
1	4	0	0.05	46	0	0.67
2	25	14	0.50	50	1	0.87
3	19	17	0.27	39	2	0.86
4	2	5	0.09	46	6	0.80
5	28	17	0.33	58	2	0.93
6	6	5	0.09	49	1	0.90
7	11	5	0.22	49	0	0.80
Total	95	63	1.55	337	12	5.83

TP: true positives; FP: false positives.

**Table 3 sensors-22-07361-t003:** *HER2* grading based on *HER2* and CEP17 counts.

SI	Quantified Area (µm${}^{2}$)	Total Quantification Time (min) Using SVM-Based Method	Total Quantification Time (min) Using Proposed Method	Proposed Nuclei Segmentation Time (min)
1	21.1 K	2.91	1.33	0.66
2	44.8 K	7.10	4.00	1.81
3	34.3 K	5.00	3.60	1.41
4	23.7 K	4.10	2.24	0.52
5	23.7 K	4.70	2.91	1.65
6	18.4 K	3.10	1.82	0.40
7	18.4 K	3.80	2.59	1.61

## Data Availability

Not applicable.

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
