# Peer review of "Singular Nuclei Segmentation for Automatic HER2 Quantification Using CISH Whole Slide Images"

_sensors, 2022, doi:10.3390/s22197361_

Round 1

Reviewer 1 Report

This article is clear and well written. The results presented are interesting and essential for optimal choice of cell quantification methods in different experimental paradigms. However the discussion section should be improved for better analysis of the methodological diversity described in recent literature. It would help to emphasized the advantages of the proposed U-net-based singular nuclei segmentation method for automatic HER2 quantification. The article could be published after this minor improvement. 

Author Response

This article is clear and well written. The results presented are interesting and essential for optimal choice of cell quantification methods in different experimental paradigms.

Point 1: However, the discussion section should be improved for better analysis of the methodological diversity described in recent literature. It would help to emphasized the advantages of the proposed U-net-based singular nuclei segmentation method for automatic HER2 quantification. The article could be published after this minor improvement. 

Response 1: Thank you very much for the comment. The discussion has been rewritten in the revised manuscript to explain the diversity and advantages of using proposed method for HER2 quantification.

The following sentences have been added to the discussion:

“The proposed nuclei segmentation method is robust against the stain variation and multiple scanners. Moreover, this method yielded higher concordance for the automatic HER2 quantification using CISH when compared with pathologists’ FISH and CISH counts. On top of that, the method is applicable for HER2 quantification using FISH. One significant benefit of having a nuclei segmentation method like the proposed method, which works for both FISH and CISH, is that it allows laboratories to freely choose the option based on their convenience.”

“The proposed method is less parameterized and has been found to be effective for a variety of scanner and staining conditions. “

“Our method not only improved segmentation performance but also ensured its clinical relevance by evaluating its results by experts and combining it with quantification methods.”

“The proposed U-net based nuclei detection method works well with limited training data. This method demonstrated high reliability when trained on a limited dataset. This is an important feature, especially for critical clinical applications where large image samples are difficult to obtain. More training data would improve the accuracy as more nuclei features could be obtained but the current performance of the model was sufficient for quantifying the limited number of nuclei for HER2 assessment. Another notable aspect of the proposed method is that it has been tested and found to be effective for multi-modal images. Furthermore, its efficacy was demonstrated through integration and demonstration with both HER2 quantification systems, CISH and FISH.”

“The proposed nuclei segmentation method was demonstrated with the HER2 quantification using a personal notebook without a GPU which took a maximum of 4 minutes per case. This is another advantage of the proposed method as it is impractical to allocate advanced computing resources in hospitals.”

Reviewer 2 Report

This paper explores the deep learning, U-net-based singular nuclei segmentation approach for automatic HER2 quantification using CISH WSI for breast cancer detection. The text is clearly written. The authors describe in detail and with interesting scientific details the experiment they performed. Taking into account, that previously, a method was developed for automatic HER2 quantification that uses SVM, the new approach outperformed the previous one in terms of efficacy.

Still, the following remarks should be considered:

1.    There is no information regarding the data availability.

2.    If the study had patient selection criteria, the type of ethical approval for the use of breast cancer samples, and the patient consent in accordance with relevant guidelines and regulations.

3.    The code availability and the software the researchers have used.

4.    The paper needs an improvement of the introduction part with new scientific reports of the same topic (i.e. deep learning for morphological features in breast cancer for the last 2 years) which will also improve the references. Also a motivation for the use of U-net in relation with the already existing deep leaning methods is needed.

Author Response

This paper explores the deep learning, U-net-based singular nuclei segmentation approach for automatic HER2 quantification using CISH WSI for breast cancer detection. The text is clearly written. The authors describe in detail and with interesting scientific details the experiment they performed. Taking into account, that previously, a method was developed for automatic HER2 quantification that uses SVM, the new approach outperformed the previous one in terms of efficacy. Still, the following remarks should be considered: 

Point 1: There is no information regarding the data availability.

Response 1: Thank you for the comment. Due to confidentiality reasons, currently it is not possible to share the dataset publicly. Though the specimens were de-identified. We have added this information in section 2.1 Dataset of the revised manuscript as:

“In this experiment, we have used 32 breast cancer CISH WSI specimens, selected randomly. The CISH WSIs were scanned using 3dhistech WSI scanner with 40x objective lens (NA 0.95) which provided an image resolution of 0.13 μm/pixel. The specimens were de-identified and didn't contain any details of the patient. However, for the confidentiality reasons, the dataset cannot be made public currently.”

Point 2: If the study had patient selection criteria, the type of ethical approval for the use of breast cancer samples, and the patient consent in accordance with relevant guidelines and regulations.

Response 2: Thank you very much for the comment. In this experiment, we selected the specimens randomly from the patients who were diagnosed with breast cancer. The data is approved by patient for research only when it is de-identified and don’t contain any details of the patient. In the revised manuscript we have included a new section as 2.1 Dataset containing the details of dataset.

Point 3: The code availability and the software the researchers have used.

Response 3: The codes developed for this research will be uploaded soon in GitHub.

Point 4: The paper needs an improvement of the introduction part with new scientific reports of the same topic (i.e. deep learning for morphological features in breast cancer for the last 2 years) which will also improve the references. Also a motivation for the use of U-net in relation with the already existing deep leaning methods is needed.

Response 4: Thank you very much for the comment. In the revised manuscript, a review of the related works is included in Section 2 as “Literature Review” to improve the introduction and the references. The motivation for using U-net has been explained in the last paragraph of Section 2 as:

“The approaches utilized for nuclei segmentation can be divided into three groups: 1) using image analysis tools such as ImageJ, CellProfiler, 3dhistech CaseViewer and others; 2) using traditional machine learning such as SVM and Random Forest and 3) using deep learning such as U-net and other pixel-wise classification methods. Among the three approaches, deep learning has achieved higher accuracy and reliability whereas U-net based methods are leading the way. U-net based pixel-wise segmentation is found most effective, fast and achieved the state of the art [21]. This motivated us to utilize the U-net based method for singular nuclei segmentation from CISH slides.”

  1. Caicedo, J.C.; Roth, J.; Goodman, A.; Becker, T.; Karhohs, K.W.; Broisin, M.; Molnar, C.; McQuin, C.; Singh, S.; Theis, F.J.; et al. Evaluation of deep learning strategies for nucleus segmentation in fluorescence images. Cytometry Part A 2019, 95, 952–965.

Reviewer 3 Report

The research topic is great and interesting, but the manuscript needs to address some major comments that I have below:

1.      The discussion needs to be more elaborated. It doesn’t talk a lot about the recent literature reports. Moreover, it reads like a conclusion rather than a discussion. The authors need to include more reports about FISH and CISH referring to previous reports and comparing them with the current study.

2.      The authors should cite the references for the previous work as described in the discussion. Lines 297-299 and 312-313.

3.      Figures 12 and 13 don’t seem correct. Maybe change the axes’ titles?

4.      None of the microscopic images have a scale bar in them. Also, the authors should mention in the materials and methods section the microscope they have used for this study.

Author Response

The research topic is great and interesting, but the manuscript needs to address some major comments that I have below:

Point 1: The discussion needs to be more elaborated. It doesn’t talk a lot about the recent literature reports. Moreover, it reads like a conclusion rather than a discussion. The authors need to include more reports about FISH and CISH referring to previous reports and comparing them with the current study.

Response 1: Thank you very much for the comment. In the revised manuscript the discussion section has been re-written and the results of the proposed method were compared with the selected existing methods. 

Point 2: The authors should cite the references for the previous work as described in the discussion. Lines 297-299 and 312-313.

Response 2: Appropriate references are included in the discussion.

Point 3: Figures 12 and 13 don’t seem correct. Maybe change the axes’ titles?

Response 3: Thanks for the comment. We have updated the Figures. In Figure 12 and 13 the x axes show the number of patient cases, we used for the comparison. The y axes show the HER2-to-CEP17 ration and average HER2 copy number per nucleus for the cases in Figure 12 and 13, respectively. Automated CISH (Proposed U-net) and Automated FISH (Proposed U-net) utilized the proposed U-net based nuclei segmentation, thus achieved the higher concordance compared to the Automated CISH (SVM) with the Manual FISH results, which is the clinical ground truth. Automated CISH (SVM) is previously proposed method.

Point 4:  None of the microscopic images have a scale bar in them. Also, the authors should mention in the materials and methods section the microscope they have used for this study.

Response 4: Thanks for you very much for the comment. The scale bars are added with the images of Figure 7 and Figure 11, in the revised manuscript. In section 2.1 Dataset under the materials and methods section, the details of the imaging device are given as:

“The CISH WSIs were scanned using 3dhistech WSI scanner with 40x objective lens (NA 0.95) which provided an image resolution of 0.13 μm/pixel.”